# Artificial Intelligence and 3D Reconstruction in Complex Hepato-Pancreato-Biliary (HPB) Surgery: A Comprehensive Review of the Literature

**DOI:** 10.3390/jpm15120610

**Published:** 2025-12-08

**Authors:** Andreas Panagakis, Ioannis Katsaros, Maria Sotiropoulou, Adam Mylonakis, Markos Despotidis, Aristeidis Sourgiadakis, Panagiotis Sakarellos, Stylianos Kapiris, Chrysovalantis Vergadis, Dimitrios Schizas, Evangelos Felekouras, Michail Vailas

**Affiliations:** 1First Department of Surgery, National and Kapodistrian University of Athens, Laikon General Hospital, 11527 Athens, Greece; andreaspanagakis@yahoo.com (A.P.); ioankats@med.uoa.gr (I.K.); adam.mylonakis@gmail.com (A.M.); markosd1995@yahoo.gr (M.D.); aristeidis.sou@gmail.com (A.S.); panagiotissakarellos@gmail.com (P.S.); schizasad@gmail.com (D.S.); felek@med.uoa.gr (E.F.); mike_vailas@yahoo.com (M.V.); 2Third Department of Surgery, Evaggelismos General Hospital of Athens, 10676 Athens, Greece; marosotiropoulou@gmail.com (M.S.); stkapiris@hotmail.com (S.K.); 3Department of Interventional Radiology, Laikon General Hospital, 11527 Athens, Greece; valvergadis@gmail.com

**Keywords:** artificial intelligence, AI, 3D reconstruction, virtual reality, HPB surgery, liver, pancreas, biliary

## Abstract

**Background:** The management of complex hepato-pancreato-biliary (HPB) pathologies demands exceptional surgical precision. Traditional two-dimensional imaging has limitations in depicting intricate anatomical relationships, potentially complicating preoperative planning. This review explores the synergistic application of three-dimensional (3D) reconstruction and artificial intelligence (AI) to support surgical decision-making in complex HPB cases. **Methods:** This narrative review synthesized the existing literature on the applications, benefits, limitations, and implementation challenges of 3D reconstruction and AI technologies in HPB surgery. **Results:** The literature suggests that 3D reconstruction provides patient-specific, interactive models that significantly improve surgeons’ understanding of tumor resectability and vascular anatomy, contributing to reduced operative time and blood loss. Building upon this, AI algorithms can automate image segmentation for 3D modeling, enhance diagnostic accuracy, and offer predictive analytics for postoperative complications, such as liver failure. By analyzing large datasets, AI can identify subtle risk factors to guide clinical decision-making. **Conclusions:** The convergence of 3D visualization and AI-driven analytics is contributing to an emerging paradigm shift in HPB surgery. This combination may foster a more personalized, precise, and data-informed surgical approach, particularly in anatomically complex or high-risk cases. However, current evidence is heterogeneous and largely observational, underscoring the need for prospective multicenter validation before routine implementation.

## 1. Introduction

Hepato-Pancreato-Biliary (HPB) surgery stands as one of the most demanding domains in surgical oncology [1,2]. It is characterized by its technical complexity, high perioperative morbidity and mortality risk, and complicated vasculo-biliary anatomy with significant inter-patient variability. Even small tumors can abut critical structures like the superior mesenteric vein (SMV), portal vein, or hepatic artery, challenging the feasibility of achieving a radical oncologic (R0) resection. This surgical challenge is compounded by the necessity to preserve an adequate future liver remnant (FLR) or a viable pancreatic remnant to prevent endocrine and exocrine insufficiency, which is a critical factor for postoperative outcomes and a core dilemma in operative planning. Complex HPB surgery encompasses procedures with increased anatomic and technical demands, including major hepatectomy (resection of three or more segments), resections requiring vascular reconstruction, operations for hilar cholangiocarcinoma, resections for borderline-resectable or locally advanced pancreatic ductal adenocarcinoma (PDAC), and living donor liver transplantation. These scenarios are characterized by narrow oncologic margins, a high risk of postoperative organ failure, and a need for meticulous preoperative planning.

For decades, conventional imaging, primarily multiphasic Computed Tomography (CT) and Magnetic Resonance Imaging (MRI), has been the backbone of preoperative planning. These methods are highly effective for diagnosis and staging, yet they display inherent limitations, as they present three-dimensional anatomy as a series of two-dimensional slices [3,4]. This forces the surgeon to perform a mental reconstruction, a cognitive process that is inherently subjective and can lead to misinterpretations of tumor boundaries, vascular encasement, and anatomical variations, even for experienced clinicians.

Three-dimensional (3D) reconstruction offers a significant step forward. By processing thin-slice imaging data, this technology generates patient-specific volumetric models that reproduce anatomy with high fidelity [5]. Surgeons can interactively manipulate these models, rotating, zooming, and making structures transparent, to gain a unique understanding of the spatial relationships between tumors, vessels, and bile ducts. The applications are highly practical: 3D models clarify vascular involvement in pancreatic tumors infiltrating the SMV/SMA, enable precise calculation of the FLR, and improve donor safety in living donor liver transplantation by depicting small anatomical branches. Crucially, this enhanced understanding is not merely academic; it has a direct clinical impact. It is reported that insights from 3D models can alter the planned surgical strategy in up to a third of complex cases, fostering more precise resections and reducing intraoperative surprises [6].

Artificial intelligence (AI) further augments these capabilities by addressing key workflow bottlenecks and adding a predictive dimension to surgical planning [7]. Its most immediate application is the automated segmentation of imaging data [8,9]. Using deep learning algorithms, this once-laborious manual process can be completed rapidly and consistently, making 3D modeling a clinical reality [10]. Beyond image processing, AI is moving into the predictive domain, with algorithms trained to estimate the risk of specific postoperative complications, like post-hepatectomy liver failure or a clinically relevant pancreatic fistula. This allows for proactive risk stratification and personalized perioperative management. The synergy between 3D visualization and AI, thus, creates a continuous digital thread that extends into the domains of surgical training and real-time surgical execution. Virtual reality (VR) platforms immerse surgeons in patient-specific simulations to rehearse complex procedures and shorten the learning curve, while intraoperative augmented reality (AR) overlays provide a real-time “roadmap” by projecting the 3D model onto the live surgical field [1].

Several narrative and systematic reviews have recently examined 3D reconstruction, AI, and VR/AR in HPB surgery. Building on this body of work, the present review aims to (i) provide an integrated overview of how 3D reconstruction and AI are applied across the full perioperative continuum in complex HPB surgery; (ii) critically appraise the quality and heterogeneity of the available evidence; and (iii) highlight implementation challenges and future research directions, including VR/AR as translational extensions of 3D and AI. The rationale for this review lies in the increasing complexity of cases, the well-recognized limitations of conventional imaging, and the mounting evidence that digital innovations are driving a paradigm shift toward a more precise and personalized standard of care in HPB surgery.

## 2. Methods

This study is a narrative review of the literature. A comprehensive search was conducted using PubMed, Scopus, and Google Scholar for articles published up to 1 October 2025. Search strategy for these databases was (“three-dimensional” OR “3D reconstruction” OR “3D modeling”) AND (“artificial intelligence” OR “machine learning” OR “deep learning”) AND (“hepato-pancreato-biliary” OR “HPB” OR “liver surgery” OR “hepatectomy” OR “pancreatic surgery” OR “pancreaticoduodenectomy”). Google Scholar was used complementarily for each query, in order to find studies not indexed in the abovementioned databases. Priority was given to English-language clinical studies, systematic reviews, and relevant technical papers. Reference lists of included studies were also screened to identify additional sources. References were imported into a reference manager and duplicates removed via automated and manual checks. Case reports were included only when they provided unique or illustrative insights into the practical application of 3D or AI technologies. Given the narrative design, no formal risk-of-bias tool was applied; instead, evidence quality is discussed qualitatively in later sections.

## 3. Preoperative Planning

### 3.1. Technical Foundation and Advantages over Conventional 2D Imaging

Preoperative imaging is the foundation upon which three-dimensional (3D) reconstructions are built. Reliable three-dimensional (3D) reconstruction relies on high-resolution preoperative imaging, ideally with slices <1 mm [2,3]. Multiphasic contrast-enhanced MDCT is the gold standard, providing essential arterial, portal venous, and delayed phases to map tumor–vessel relationships (celiac axis, SMA, portal vein, etc.). MRI (often with MRCP) complements CT by enhancing visualization of the biliary and pancreatic ductal systems [1].

Data is imported into specialized software for segmentation—the process of isolating and defining the pancreas, vessels, and biliary tract. This results in a patient-specific 3D model that can be manipulated virtually. Increasingly, these models are used to create 3D-printed models or are projected in augmented reality (AR) for intraoperative guidance [3,6]. This combination of CT, MRI, segmentation software, and—increasingly—printing or AR visualization forms the technical backbone of modern 3D surgical planning

Traditional 2D imaging requires surgeons to mentally reconstruct the anatomy from stacks of axial slices, a subjective process dependent on individual experience that can lead to subtle findings being overlooked. Three-dimensional reconstruction resolves this issue by providing an objective, interactive map of the operative field [6]. The primary benefit is enhanced anatomical clarity, as 3D models intuitively display the complex spatial relationships between the tumor and critical structures, such as highlighting encasement or abutment of the SMA and portal vein, and more easily identifying anatomical variants [2,6]. Three-dimensional volume rendering permits precise tumor assessment, with measured tumor volumes correlating more closely with pathological specimens than conventional CT measurements [2]. Furthermore, 3D models significantly improve resection planning. They allow surgeons to virtually simulate resection planes, anticipate the need for vascular reconstruction, and employ volumetric planning (like future liver remnant calculation) to balance oncologic clearance and functional preservation [[11],,[12]].

Finally, these models carry educational and communicative value. For trainees, they provide an immersive understanding of anatomical complexity that surpasses textbook diagrams [1,13]. For patients, printed or virtual reconstructions offer a tangible way to appreciate the planned operation, facilitating informed consent and shared decision-making [14]. This shift toward 3D planning, increasingly automated by AI, represents a vital transition toward a digital era of complex surgical execution.

### 3.2. Impact on Surgical Strategy

The strongest argument for integrating 3D reconstruction into preoperative planning is its demonstrated ability to alter surgical strategy. Several clinical series have reported changes in operative planning in 20–30% of cases once 3D reconstructions were reviewed. These changes typically involve revised judgments about vascular invasion, resectability, and the extent of resection required. For example, in a study of pancreatoduodenectomies, surgical plans were modified in more than one-fifth of patients after 3D evaluation, with the most frequent adjustments relating to anticipated venous resection or reconstruction [2]. In borderline resectable pancreatic ductal adenocarcinoma (BR-PDAC), where accurate assessment of venous and arterial involvement determines operability, 3D reconstructions have proven particularly useful [6,11]. They allow surgeons to visualize tumor abutment of the SMV–portal vein confluence or encasement of arterial branches in a way that 2D slices cannot, as shown in Figure 1. This not only aids in selecting candidates for upfront surgery versus neoadjuvant therapy but also prepares the surgical team for complex vascular procedures if required.

The influence of 3D planning is equally evident in liver surgery, where its adoption is more advanced [4,15]. 3D-based volumetry leads to shorter operative times, reduced intraoperative blood loss, and decreased transfusion requirements compared with conventional imaging [4,15]. Moreover, early evidence suggests that 3D planning contributes to higher rates of R0 resections and fewer intraoperative surprises. Another relevant impact concerns operative efficiency. By clarifying vascular anatomy and resection planes before the incision, 3D planning can shorten the learning curve for young surgeons and provide confidence for senior surgeons facing complex cases [4]. Beyond individual benefit, these tools may help standardize surgical planning across institutions by reducing reliance on subjective mental reconstruction.

In summary, 3D reconstruction has transformed preoperative planning for complex HPB surgery from a largely interpretative exercise into a reproducible, data-driven process. By ensuring high-quality imaging, accurately segmenting relevant anatomy, and delivering models that improve clarity, precision, and strategy, these technologies directly influence patient outcomes. They allow surgeons not only to plan the procedure with greater certainty but also to anticipate difficulties, avoid unnecessary exploratory laparotomies, and individualize surgical strategies. As 3D software becomes more automated and as AR/VR platforms mature, these reconstructions are poised to become standard practice, not an optional adjunct, in the preoperative evaluation of pancreatic cancer.

## 4. Intraoperative Applications of 3D Reconstruction in Liver and Pancreatic Surgery

The role of three-dimensional (3D) reconstruction in HPB surgery has steadily expanded over the last decade [15]. Initially developed as a planning tool to understand complex anatomy before surgery, it is now moving into the operating room itself. Surgeons can rely on interactive 3D visualizations, augmented reality (AR), and even tangible printed organ models while operating. These technologies provide a dynamic way to interpret anatomy and pathology. This is especially relevant in liver and pancreatic operations, where the close relationship between tumors, vessels, and ducts leaves very little margin for error. The goal is not only to improve precision, but also to lower the risks of complications and improve the patient’s recovery afterwards.

### 4.1. Intraoperative Applications in Pancreatic Surgery

Pancreatic surgery often involves challenging operations, such as pancreatoduodenectomy (PD), involving dissection around the superior mesenteric vein, the portal vein, and the celiac axis, where a small error in vascular dissection can have catastrophic consequences. Miyamoto et al. studied 117 patients undergoing PD and reported that those with 3D simulation had significantly less blood loss compared to conventional planning [16], an important finding given that blood loss and transfusions are well-known risk factors for postoperative morbidity.

Isolated reports also highlight the ability of 3D reconstructions to reveal details that traditional imaging may miss (Figure 2). Templin et al. described a case in which interactive 3D vascular reconstruction revealed a rare Bühler’s anastomosis, not visible on standard CT [7]. This information, confirmed during surgery, changed the operative strategy and likely prevented serious complications. Vicente et al. went further, showing that 3D vascular models achieved nearly 100% sensitivity and specificity in predicting vascular invasion in pancreatic cancer, outperforming CT and MRI [3]. This level of accuracy allows for better planning of vascular resections and helps avoid unnecessary or unsafe procedures.

Augmented reality further extends these advantages. Tang et al. reported AR-assisted PD procedures that required venous resection [17]. By projecting 3D reconstructions onto the operative field, they improved the surgeon’s orientation and helped to achieve negative resection margins. Collectively, these reports suggest that 3D support during pancreatic surgery does more than enhance visualization; it actively contributes to safer dissection, reduces the risk of vascular injury, and lowers the likelihood of complications such as postoperative pancreatic fistula.

### 4.2. Intraoperative Applications in Liver Surgery

Liver surgery is one of the main areas where 3D reconstruction has demonstrated its potential. During major resections, a precise understanding of the vascular inflow, venous drainage, and bile duct anatomy is essential, as well as an accurate calculation of the future liver remnant (FLR), which is crucial (Figure 3). Conventional CT can provide this information but often requires the surgeon to mentally reconstruct 3D relationships from 2D slices. Cotsoglou et al. carried out a multicentric international survey and demonstrated that 3D reconstructions significantly improved the recognition of tumor–vessel relations compared with standard CT [4]. Surgeons reported that this information reduced intraoperative uncertainty and allowed for more confident intraoperative decisions.

Another breakthrough comes from AR-guided approaches. Wang and colleagues combined 3D reconstruction with intraoperative fluorescence guidance in laparoscopic hepatectomy [18]. They found shorter operative times, reduced blood loss, and even lower postoperative stress responses, suggesting that intraoperative navigation not only makes the surgery more efficient but also reduces systemic strain on the patient. Rossi et al., in a systematic review, emphasized the role of patient-specific 3D-printed liver models [19]. These models helped surgeons anticipate variations in vascular and biliary anatomy, reducing the risk of unexpected injury during the procedure.

This evidence is also supported by pooled analyses. For instance, Zeng et al. included more than 1600 patients in a meta-analysis and confirmed that 3D-assisted hepatectomy is associated with reduced operative time, less bleeding, fewer complications, lower recurrence rates, and improved survival [20]. These data suggest that the intraoperative use of 3D reconstructions is a real clinical tool with a meaningful impact.

### 4.3. Postoperative Outcomes and Clinical Impact

The influence of intraoperative 3D support does not end when the operation is finished. Its benefits extend into the postoperative course. In liver surgery, reduced bleeding and shorter operative times translate into fewer bile leaks, lower risk of postoperative liver failure, and shorter hospital stays [21]. In pancreatic surgery, accurate recognition of vascular involvement allows for safer resections, which reduces intraoperative injury and decreases the incidence of postoperative pancreatic fistula [22]. Importantly, by helping to achieve R0 resections, intraoperative 3D guidance may also lower recurrence rates and support better long-term survival [23]. Taken together, surveys, retrospective cohorts, prospective series, and meta-analyses suggest a favorable signal that intraoperative 3D support may improve both technical precision and selected clinical outcomes. However, most evidence is observational and heterogeneous, so further prospective validation is required before firm conclusions can be drawn [13]. Looking ahead, integration with AI-based segmentation, robotics, and immersive AR platforms will likely make intraoperative 3D guidance an everyday part of surgical practice.

## 5. Artificial Intelligence and Virtual Reality in HPB Surgery

Artificial intelligence (AI) and virtual reality (VR) are now appearing increasingly in HPB surgery. They rely on the progress of 3D reconstruction but go a step further, as they not only help in showing anatomy before an operation, but also create tools that predict risks, allowing rehearsal before surgery and supporting the surgeon during the operation.

AI predictive models are mainly used for risk estimation. By analyzing imaging data together with clinical variables, machine learning can identify patients who are more likely to develop complications. In pancreatic surgery, one of the biggest issues is postoperative pancreatic fistula. Preliminary studies suggest that AI models may outperform traditional risk scores, although external validation is still limited. Vicente et al. showed that 3D-based modeling was more accurate than CT or MRI in detecting vascular infiltration, which directly influenced the surgical plan [3]. In liver surgery, algorithms that process 3D reconstructions have been used to predict whether the future liver remnant is sufficient, helping to avoid postoperative liver failure. Meta-analyses such as that of Zeng et al. have confirmed that patients operated on 3D support have fewer complications and better survival [20].

VR for training and education is another rapidly growing field. Hepatic and pancreatic operations are among the most demanding in surgical oncology, and VR tools provide a safe way to train [13]. Based on patient-specific 3D models, VR systems allow the surgeon to “walk through” the anatomy before actually entering the operating room. Studies such as Oshiro et al. and Rossi et al. described how VR or 3D-printed models improved spatial understanding, especially of vascular and biliary variations. More importantly, these tools are not only for residents, but experienced surgeons can also use them to test different strategies or rehearse rare scenarios [19,24].

Intraoperative VR guidance closes the circle. Real-time overlays of 3D anatomy onto the surgical field are already used in both liver and pancreatic operations. Tang et al. reported the use of VR during pancreatoduodenectomy with vascular resection; the 3D overlay made dissection safer and helped to achieve negative margins [17]. In liver surgery, Wang et al. combined VR navigation with fluorescence imaging, which resulted in less blood loss and shorter procedures [12]. A striking example comes from Templin et al., who showed that an unexpected vascular variant, missed on conventional CT, was identified with 3D vascular reconstruction and confirmed during surgery, preventing possible major complications [7].

These developments have tangible implications for patients. They may contribute to fewer complications such as bleeding, bile leaks, or pancreatic fistula, shorter hospital stays, and—in selected settings—better long-term disease control. VR-guided resections can improve margin status, which may reduce recurrence, while AI-based prediction supports better patient selection by identifying individuals at particularly high risk. At the same time, VR training can help ensure that younger surgeons are better prepared for complex anatomy. Nonetheless, important challenges remain. Production of patient-specific 3D/VR models can be time-consuming and costly, as Rossi et al. noted [19]. AI models require large, high-quality, and diverse datasets to achieve robust performance, which in turn demands sustained multicenter collaboration. As a result, AI, VR, and AR are best viewed as promising and rapidly maturing tools whose integration into routine HPB practice will depend on further validation, cost-effectiveness analyses, and workflow optimization.

In conclusion, these three elements form a complementary triad. AI helps to forecast complications, VR training improves preparation, and intraoperative VR supports safe execution. Together, they reduce perioperative risk and hold the promise of better outcomes for patients facing complex liver and pancreatic operations.

## 6. Current Challenges and Future Perspectives

Despite a growing body of encouraging evidence, the clinical translation of 3D reconstruction and AI in complex HPB surgery still faces significant practical and scientific hurdles. A primary obstacle is the resource-intensive nature of current workflows. Generating a patient-specific 3D model often involves time-consuming manual or semi-manual segmentation by specialized staff, limiting rapid turnaround in urgent cases and adding substantial cost [10]. Furthermore, many published studies are single-center series with relatively small cohorts and heterogeneous endpoints, which constrain the generalizability of their findings. This heterogeneity, coupled with the risk of publication bias, makes it difficult to draw definitive conclusions regarding superiority, cost-effectiveness, and optimal indications. Practical considerations such as the cost of model generation, increased workflow burden, variable production times for patient-specific 3D models, and differences in software accuracy also represent important barriers to widespread adoption.

To move from promising adjuncts to standard of care, future research must address several key questions:

1. How can segmentation and model generation be fully or nearly fully automated within real-world timeframes?

Advances in deep learning could substantially reduce manual input, but models must be robust to variable image quality and scanner protocols. Benchmarking against expert manual segmentation and validating across institutions will be essential.

2. What endpoints and datasets are needed for AI models to be trusted in HPB surgery?

Large, multicenter prospective datasets should include standardized outcome measures such as operative time, transfusion requirements, major complications (Clavien–Dindo > III), POPF, post-hepatectomy liver failure (PHLF) as defined by ISGPS/ISGLS, R0 resection rates, recurrence patterns, and survival. Transparent reporting and external validation are crucial to avoid overly optimistic performance estimates.

3. How can these tools be integrated into existing clinical workflows without increasing burden or inequity?

Implementation research is required to understand training needs, potential bottlenecks, and differences between high-volume and low-volume centers. Cost-effectiveness analyses, including the impact on length of stay, complication-related readmissions, and resource use, will help justify investment.

In parallel, ethical and regulatory considerations must be addressed. AI-based surgical tools rely on sensitive imaging and clinical data, raising concerns regarding patient privacy, data security, and ownership of algorithm-generated insights. The absence of standardized validation frameworks and regulatory pathways for surgical AI poses additional challenges. Establishing clear ethical guidelines, transparent data governance, and harmonized international regulatory standards will be essential before broad integration into daily practice.

A realistic future vision must therefore balance innovation with accountability. If adequately validated and responsibly implemented, 3D reconstruction, AI, and immersive technologies have the potential to support safer, more standardized, and more personalized HPB surgery, while ensuring that patient autonomy, privacy, and equity remain central.

## 7. Limitations

Although the available literature demonstrates encouraging results, several limitations should be acknowledged. Most of the evidence derives from case reports, small cohort studies, or single-center experiences, often with heterogeneous endpoints and a lack of randomization. The lack of standardized metrics for evaluating 3D and AI tools limits comparability across studies. Additionally, the rapid technological turnover may outpace clinical validation, while cost, data storage demands, and dependence on specialized personnel remain substantial barriers. These factors highlight the need for multicenter, prospective trials to confirm the reproducibility and long-term clinical benefits of these digital tools. Publication bias is another concern, as positive experiences with 3D and AI are more likely to be reported than neutral or negative results. Furthermore, rapid technological evolution may outpace rigorous clinical evaluation

## 8. Conclusions

Three-dimensional reconstruction, artificial intelligence, and immersive technologies such as virtual and augmented reality are steadily transforming HPB surgery. Across both liver and pancreatic procedures, these tools are associated in multiple studies with improved anatomical clarity, enhanced operative planning, and potentially better perioperative outcomes. Reported benefits include shorter operative times, reduced intraoperative blood loss, and lower rates of selected complications, with early data suggesting possible improvements in oncologic endpoints such as R0 resection rates and recurrence in specific contexts.

Their greatest promise lies in supporting a shift from a purely experience-driven process toward a more standardized and data-informed surgical workflow. By making complex anatomy more accessible, enabling individualized risk stratification, and providing new training and navigation platforms, these technologies may ultimately improve care for patients facing the most challenging HPB operations. At present, however, the evidence remains heterogeneous and largely observational. Continued efforts to generate high-quality, multicenter, and methodologically rigorous data will be essential to confirm these benefits and to define when, how, and for whom 3D, AI, and VR/AR should be integrated into routine HPB practice.

## Figures and Tables

**Figure 1 jpm-15-00610-f001:**
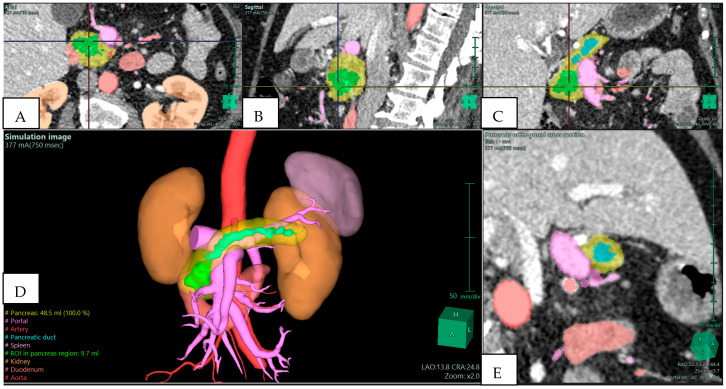
Preoperative Planning of a Pancreatic Head Adenocarcinoma with Main Duct Intraductal Papillary Mucinous Neoplasm (IPMN) in a 70-year-old Woman. Multiphase CT scan with 3D volume rendering shows the tumor (green) abutting the superior mesenteric vein (SMV)/portal vein (PV) confluence (purple). Images derived from anonymized clinical datasets from our institution. (**A**) Axial plane; (**B**) sagittal plane; (**C**) coronal plane; (**D**) 3D reconstruction—simulation image; and (**E**) pancreas orthogonal cross section. Images derived from anonymized clinical datasets from our institution.

**Figure 2 jpm-15-00610-f002:**
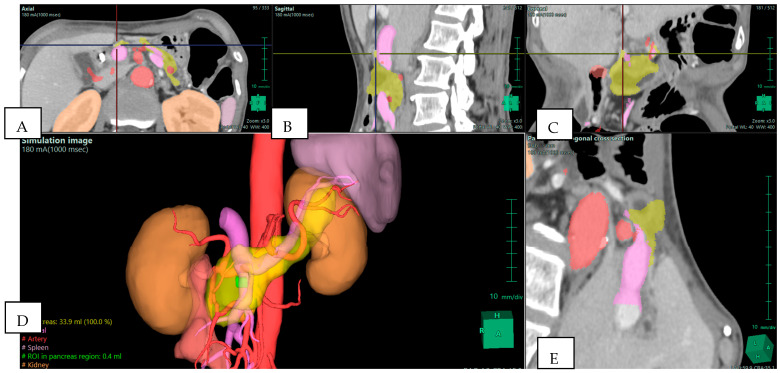
Preoperative Planning of Uncinate Process Pancreatic Adenocarcinoma (PDAC) in a 69-year-old Woman. Multiphase CT scan with 3D volume rendering shows the tumor (green) arising from the uncinate process and abutting the superior mesenteric vein (SMV)/portal vein (PV) confluence (purple). Note the presence of an aberrant right hepatic artery originating from the superior mesenteric artery (SMA) (red). (**A**) Axial plane; (**B**) sagittal plane; (**C**) coronal plane; (**D**) 3D reconstruction—simulation image; and (**E**) pancreas orthogonal cross section. Images derived from anonymized clinical datasets from our institution.

**Figure 3 jpm-15-00610-f003:**
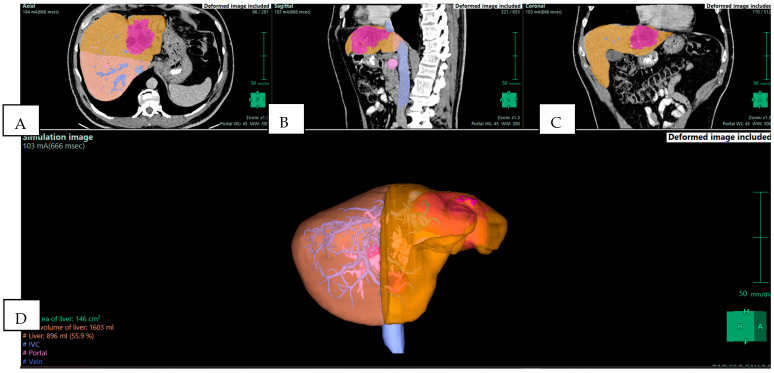
Preoperative planning of a mass-forming intrahepatic cholangiocarcinoma (iCCA) in the left lobe of the liver in a 57-year-old man. The tumor (pink) is in the left lobe (yellow). The future liver remnant (orange) is calculated for the right lobe and is 55.9% of the total liver volume (TLV), indicating that a left hepatectomy is feasible. (**A**) Axial plane; (**B**) sagittal plane; (**C**) coronal plane; and (**D**) 3D reconstruction—simulation image. Images derived from anonymized clinical datasets from our institution.

## Data Availability

No new data were created or analyzed in this study. Data sharing is not applicable to this article.

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
