# Peer review of "Artificial Intelligence and 3D Reconstruction in Complex Hepato-Pancreato-Biliary (HPB) Surgery: A Comprehensive Review of the Literature"

_jpm, 2025, doi:10.3390/jpm15120610_

Round 1
Reviewer 1 Report
Comments and Suggestions for Authors
Dear Editor, dear Authors,
Thank you for the opportunity to review this manuscript. I appreciate the authors’ efforts to conduct this narrative review examining the emerging role of three-dimensional (3D) reconstruction and artificial intelligence (AI) in complex hepato-pancreato-biliary (HPB) surgery. The manuscript synthesizes current evidence on how these digital technologies may improve preoperative planning, intraoperative navigation, risk prediction, and surgical training. The review highlights that 3D reconstruction provides highly accurate, patient-specific anatomical models derived from thin-slice CT/MRI imaging, enabling precise evaluation of tumor–vessel relationships, anatomical variants, and resectability. These models frequently lead to modified surgical strategies in complex HPB cases and are associated with reduced operative time, blood loss, and intraoperative uncertainty. AI algorithms further enhance surgical workflows by automating image segmentation, improving diagnostic accuracy regarding vascular invasion, and predicting postoperative complications such as post-hepatectomy liver failure or clinically relevant pancreatic fistula. Virtual reality (VR) and augmented reality (AR) applications support both surgical rehearsal and real-time intraoperative guidance, complementing the benefits of 3D and AI technologies. Overall, the review concludes that digital tools are gradually transforming HPB surgery into a more precise, personalized, and data-driven discipline.
This review stands out for its well-structured synthesis of the literature and its clear attempt to integrate evidence across pancreatic and liver surgery. The authors convincingly suggest that these technologies improve anatomical clarity, enhance surgical precision, and contribute to better perioperative outcomes—including reduced operative time, lower blood loss, fewer complications, and improved R0 resection rates.
Despite its strengths, the manuscript presents several important limitations:
- Although this manuscript is a narrative review, it would still benefit from a clearer description of the literature search strategy and from briefly outlining the criteria used to select the studies discussed. While full systematic review methodology is not required, providing at least a concise overview of the search process—and optionally a simple flowchart—would enhance transparency and allow readers to better understand how the included evidence was identified.
- Important practical aspects—such as cost, workflow burden, time required for model generation, and software accuracy—can be addressed.
- Inconsistencies appear in the administrative statements: the manuscript states both that informed consent was obtained and that informed consent is not applicable (line 400-402). These contradictory declarations can be inappropriate for a narrative review and should be reevaluated.
- Figure 2 is introduced visually before it is cited in the text.
- The source of the figures and CT scans should be clearly stated.
- When multiple visual elements are presented within the same figure, each panel should be labeled (e.g., A, B, C, D) and described individually in the figure legend to clarify their distinct purpose and differences.
- The number of references is relatively limited for a manuscript that aims to provide a comprehensive state-of-the-art overview. Expanding the reference list would strengthen the manuscript’s credibility and perceived depth.
I hope these comments prove constructive and contribute to improving the quality of the manuscript. The suggestions provided are not mandatory; the authors are encouraged to implement those they find most appropriate and valuable.
I appreciate the work and effort invested in preparing this review.
With respect and consideration,
Reviewer
Author Response
Comment 1:
Dear Editor, dear Authors,
Thank you for the opportunity to review this manuscript. I appreciate the authors’ efforts to conduct this narrative review examining the emerging role of three-dimensional (3D) reconstruction and artificial intelligence (AI) in complex hepato-pancreato-biliary (HPB) surgery. The manuscript synthesizes current evidence on how these digital technologies may improve preoperative planning, intraoperative navigation, risk prediction, and surgical training. The review highlights that 3D reconstruction provides highly accurate, patient-specific anatomical models derived from thin-slice CT/MRI imaging, enabling precise evaluation of tumor–vessel relationships, anatomical variants, and resectability. These models frequently lead to modified surgical strategies in complex HPB cases and are associated with reduced operative time, blood loss, and intraoperative uncertainty. AI algorithms further enhance surgical workflows by automating image segmentation, improving diagnostic accuracy regarding vascular invasion, and predicting postoperative complications such as post-hepatectomy liver failure or clinically relevant pancreatic fistula. Virtual reality (VR) and augmented reality (AR) applications support both surgical rehearsal and real-time intraoperative guidance, complementing the benefits of 3D and AI technologies. Overall, the review concludes that digital tools are gradually transforming HPB surgery into a more precise, personalized, and data-driven discipline.
This review stands out for its well-structured synthesis of the literature and its clear attempt to integrate evidence across pancreatic and liver surgery. The authors convincingly suggest that these technologies improve anatomical clarity, enhance surgical precision, and contribute to better perioperative outcomes—including reduced operative time, lower blood loss, fewer complications, and improved R0 resection rates.
Despite its strengths, the manuscript presents several important limitations:
Although this manuscript is a narrative review, it would still benefit from a clearer description of the literature search strategy and from briefly outlining the criteria used to select the studies discussed. While full systematic review methodology is not required, providing at least a concise overview of the search process—and optionally a simple flowchart—would enhance transparency and allow readers to better understand how the included evidence was identified.
Response 1: Thank you for your valuable comment. We included in the methods section of our manuscript a detailed search strategy for our study. This is a narrative review and a critical appraisal of the available literature was done. As a result a flowchart is not applicable.
Comment 2:
Important practical aspects—such as cost, workflow burden, time required for model generation, and software accuracy—can be addressed.
Response 2:
Thank you for your feedback. Indeed, these aspects are important for the future implementation of these technologies. We briefly addressed them in revised sections 5 and 6.
Comment 3:
Inconsistencies appear in the administrative statements: the manuscript states both that informed consent was obtained and that informed consent is not applicable (line 400-402). These contradictory declarations can be inappropriate for a narrative review and should be reevaluated.
Response 3: Thank you for your comment. Inconsistencies are addressed accordingly.
Comment 4: Figure 2 is introduced visually before it is cited in the text.
Response 4: Thank you for your comment. Figure 2 is moved after the text citation.
Comment 5: The source of the figures and CT scans should be clearly stated.
Response 5: Thank you for your comment. Figures come from analysis of our patients’ scans.
Comment 6: When multiple visual elements are presented within the same figure, each panel should be labeled (e.g., A, B, C, D) and described individually in the figure legend to clarify their distinct purpose and differences.
Response 6: Thank you for your input. Labels are added in all figures and described accordingly in figure legends.
Comment 7: The number of references is relatively limited for a manuscript that aims to provide a comprehensive state-of-the-art overview. Expanding the reference list would strengthen the manuscript’s credibility and perceived depth.
Response 7: Thank you for your comment. We understand your concern, but our study is a comprehensive review of existing literature. We opted to discuss the main studies and reviews on the subject, while also stating our views on current challenges and future directions.
Reviewer 2 Report
Comments and Suggestions for Authors
Comments:
1. The manuscript is titled as a “comprehensive review of the literature”, and the Introduction states that it aims to provide an overview of how 3D reconstruction and AI are reshaping complex HPB management. However, several narrative and systematic reviews on 3D, AI, VR/AR in HPB and hepatobiliary surgery have been published recently (some of which you already cite in the References section, e.g. Boul-Atarass et al., Zhang et al., Rossi et al.).Explicitly state in the Introduction and the first paragraph of the Discussion what is new here compared with previous reviews.
2. Methods lack reproducibility (no database-specific Boolean strings, interfaces, date per database, or deduplication steps; Google Scholar included without capture method). Revise by: supplying full copy-pastable strategies for each database, platforms used, last-search date per database, and a deduplication workflow; justify Google Scholar use.
3. The manuscript repeatedly uses strong phrases such as “consistently demonstrated benefits”, “real clinical tool with meaningful impact”, and “are steadily transforming HPB surgery”, even though most cited data are small, single-center cohorts and retrospective studies.Please temper the language (e.g., “available evidence suggests…”, “early data indicate…”) and clearly distinguish between hypothesis-generating findings and robust, confirmatory data.
4. Scope definition: what qualifies as “complex HPB surgery”?The manuscript focuses on complex HPB cases but does not explicitly define what is meant by “complex” (major hepatectomy, vascular resections, living donor transplantation, borderline-resectable pancreatic cancer, etc.).
5. Sections 3.1 and 3.2 explain the technical requirements, advantages of 3D over 2D imaging, and evolution toward interactive models and AI-assisted segmentation.Some ideas (mental reconstruction, subjectivity of 2D slices, transition to interactive 3D) appear multiple times. Consider tightening these paragraphs, eliminating repeated phrases, and focusing more on quantitative effects (how 3D changes resection planning, FLR calculations, or resectability assessment) with concise linkage to the cited studies.
6. The review cites many encouraging results (shorter operative time, reduced blood loss, fewer complications, better survival) from surveys, retrospective cohorts, and meta-analyses. However, the Discussion remains largely descriptive. It would be useful to explicitly comment on: heterogeneity of endpoints across studies, predominance of single-center settings, limited randomized data, and potential publication bias.
7. Highlighting future research – structure the “Current Challenges and Future Perspectives” section more clearlySection 6 contains an excellent discussion of workflow, standardization, and ethical/regulatory issues. To strengthen it, consider structuring it around a few explicit key questions, for example:How can fully automated segmentation and rapid model generation be achieved in real-world timeframes? W hat endpoints and datasets are needed for AI models to be trusted in HPB surgery?
8. The English is generally good, but there are several overly long sentences and a few typographical issues (e.g., “golden standard” should be “gold standard”; “Posτoperative Outcomes” contains a stray Greek letter τ; occasional informal phrases such as “these technologies are not just an additional gadget”).I recommend a professional language editing.
9. The title emphasizes “Artificial Intelligence and 3D Reconstruction,” while the review also covers virtual reality and augmented reality in depth (Section 5).Since VR is already in the keywords, you may consider whether the title should explicitly mention VR/AR as well (optional). Alternatively, clarify early in the Introduction that the scope includes VR/AR as downstream applications of 3D and AI.
Author Response
Comment 1:
The manuscript is titled as a “comprehensive review of the literature”, and the Introduction states that it aims to provide an overview of how 3D reconstruction and AI are reshaping complex HPB management. However, several narrative and systematic reviews on 3D, AI, VR/AR in HPB and hepatobiliary surgery have been published recently (some of which you already cite in the References section, e.g. Boul-Atarass et al., Zhang et al., Rossi et al.).Explicitly state in the Introduction and the first paragraph of the Discussion what is new here compared with previous reviews.
Response 1: Thank you for your input. We added a relevant section at the Introduction of our manuscript.
Comment 2: Methods lack reproducibility (no database-specific Boolean strings, interfaces, date per database, or deduplication steps; Google Scholar included without capture method). Revise by: supplying full copy-pastable strategies for each database, platforms used, last-search date per database, and a deduplication workflow; justify Google Scholar use.
Response 2: Thank you for your comment. This is a comprehensive review of the literature and we critically discuss available evidence. Search strategy of included databases is added at the Methods section. Google Scholar was used complementarily for each query, in order to find studies not indexed in the abovementioned databases.
Comment 3: The manuscript repeatedly uses strong phrases such as “consistently demonstrated benefits”, “real clinical tool with meaningful impact”, and “are steadily transforming HPB surgery”, even though most cited data are small, single-center cohorts and retrospective studies.Please temper the language (e.g., “available evidence suggests…”, “early data indicate…”) and clearly distinguish between hypothesis-generating findings and robust, confirmatory data.
Response 3: Thank you for your valuable feedback. We reviewed the whole manuscript and tempered down the language throughout the text.
Comment 4: Scope definition: what qualifies as “complex HPB surgery”?The manuscript focuses on complex HPB cases but does not explicitly define what is meant by “complex” (major hepatectomy, vascular resections, living donor transplantation, borderline-resectable pancreatic cancer, etc.).
Response 4: Thank you for your comment. We defined complex HPB surgery at the beginning of Introduction to clarify its use throughout the manuscript.
Comment 5: Sections 3.1 and 3.2 explain the technical requirements, advantages of 3D over 2D imaging, and evolution toward interactive models and AI-assisted segmentation.Some ideas (mental reconstruction, subjectivity of 2D slices, transition to interactive 3D) appear multiple times. Consider tightening these paragraphs, eliminating repeated phrases, and focusing more on quantitative effects (how 3D changes resection planning, FLR calculations, or resectability assessment) with concise linkage to the cited studies.
Response 5: Thank you for your input. We merged theses two sections.
Comment 6: The review cites many encouraging results (shorter operative time, reduced blood loss, fewer complications, better survival) from surveys, retrospective cohorts, and meta-analyses. However, the Discussion remains largely descriptive. It would be useful to explicitly comment on: heterogeneity of endpoints across studies, predominance of single-center settings, limited randomized data, and potential publication bias.
Response 6: Thank you for comment. This is a narrative review of the literature and no risk of bias assessment of individual studies was applicable. We critically appraised each study in the manuscript.
Comment 7: Highlighting future research – structure the “Current Challenges and Future Perspectives” section more clearlySection 6 contains an excellent discussion of workflow, standardization, and ethical/regulatory issues. To strengthen it, consider structuring it around a few explicit key questions, for example:How can fully automated segmentation and rapid model generation be achieved in real-world timeframes? W hat endpoints and datasets are needed for AI models to be trusted in HPB surgery?
Response 7: Thank you for your valuable input. We added key questions in Section 6.
Comment 8: The English is generally good, but there are several overly long sentences and a few typographical issues (e.g., “golden standard” should be “gold standard”; “Posτoperative Outcomes” contains a stray Greek letter τ; occasional informal phrases such as “these technologies are not just an additional gadget”).I recommend a professional language editing.
Response 8: Thank you for your comment. A native English speaker reviewed the whole manuscript and corrected potential grammar and syntax errors.
Comment 9: The title emphasizes “Artificial Intelligence and 3D Reconstruction,” while the review also covers virtual reality and augmented reality in depth (Section 5).Since VR is already in the keywords, you may consider whether the title should explicitly mention VR/AR as well (optional). Alternatively, clarify early in the Introduction that the scope includes VR/AR as downstream applications of 3D and AI.
Response 9: Thank you for your feedback. We revised Introduction clearly stating that VR/AR are included in the scope of our manuscript as application of 3D and AI.
Round 2
Reviewer 2 Report
Comments and Suggestions for Authors
The authors have addressed all of my comments.
Comments on the Quality of English Language
The manuscript can be accepted for publication.